# Utilization of Carob Fruit as Sources of Phenolic Compounds with Antioxidant Potential: Extraction Optimization and Application in Food Models

**DOI:** 10.3390/foods9010020

**Published:** 2019-12-24

**Authors:** Vlasios Goulas, Eva Georgiou

**Affiliations:** Department of Agricultural Sciences, Biotechnology and Food Science, Cyprus University of Technology, Lemesos 3603, Cyprus; evg.georgiou@edu.cut.ac.cy

**Keywords:** antioxidant activity, carob, *Ceratonia siliqua* L., emulsion, HPLC-DPPH, lipid oxidation, polyphenols, ultrasound assisted extraction

## Abstract

The goal of this study was to explore the potential of carob extracts to act as lipid inhibitors in model food systems. First, the extraction efficacy of fourteen solvents on the phenolic and flavonoid contents as well as on the antioxidant activity was assessed. Results showed that the phenolic composition and antioxidant activity of the extracts were strongly affected by solvents. Subsequently, the antioxidant potential of the most promising extracts (water, methanol, acidic acetone, and acetone–water) against four model food systems were evaluated. The acidic acetone extract had the highest antioxidant activity (70.3 ± 5.3%) in the β-carotene-linoleic acid system, followed by the acetone–water extract (62.1 ± 4.9%). Both extracts significantly prevented the lipid oxidation in sunflower oil and cooked comminuted pork; the inhibition activity at the end of storage period was 36.7–50.5% and 17.4–24.8%, respectively. A reduction of 49.5–54.8% in the formation of dienes in the oil-in-water emulsion was also found. The inhibitory effect of methanolic and aqueous extracts was significantly lower. Qualitative and quantitative variations in extracts are responsible for this antioxidant behavior in food systems. Gallic acid, myrecetin, rutin, and catechin are the main components of the extracts while myricetin and quercetin play an essential role in the antioxidant activity, according to the biochromatograms.

## 1. Introduction

Oxidation is one of the major causes of quality deterioration in foods. Lipid oxidation induces significant loss in the nutritional value in lipid foods and emulsions, affecting their quality characteristics and safety. Therefore, the use of antioxidant additives such as butylated hydroxytoluene (E321), butylated hydroxyanisole (E320), tert-butylhydroquinone (E319), and propyl gallate (E310) to retard oxidation and peroxidation is a common practice for the food industry [1,2]. Although long-term toxicity studies have been performed for the safety of synthetic antioxidants, there is a public conviction that natural antioxidants are safer than their synthetic analogues. Thus, the food industry is seeking natural, safe, and low-cost antioxidant compounds or fractions to substitute synthetic additives [3]. In this challenge, natural sources such as plants, food by-products, and marina flora have been extensively studied.

The carob tree (*Ceratonia siliqua* L.) is mostly found in the Mediterranean Basin and is considered as an underutilized crop [4]. However, carob fruit is an essential reservoir of nutritional and pharmaceutical compounds. More specific, carob seeds are usually utilized for the production of carob bean gum (E410), which is used as thickener, stabilizer, or flavoring in the food industry and as an active ingredient or carrier molecule in pharmaceuticals [4]. Carob pulp, the main part of the fruit, is also rich in sugars (48–56%), which are extracted to produce syrups and molasses. Carob pulp has been also recognized as an excellent source of bioactive fibers [5]. Thus, many patents have been registered for the manipulation of these components of carob fruit. In contrast, the phenolic fraction of carob fruit is relatively unexploited, although significant amounts of phenolic compounds are found in carob fruit (1.2–7.0%), which is comparable to other well-known Mediterranean plant-based foods [6]. Furthermore, carob fruit contains a variety of bioactive phytochemicals from several classes [7]. In particular, carob phenolic compounds mostly comprise hydroxybenzoic acids, flavonols, flavan-3-ols, and gallotannins [4,8]. 

In the last decade, there have been some attempts to recover phenolic antioxidants from carob fruit using different extraction techniques and solvents [9,10,11]. In addition, researchers have attempted to utilize carob antioxidants to reduce lipid oxidation in cooked pork meat [12], Atlantic horse mackerel [13], and sunflower [14] as well as improve the antioxidant potential in durum wheat pasta [15] and milk beverages [16]. To the best of our knowledge, information about the phenolic composition and antioxidant potency of carob fruit is still obscure as there is no available comparative study on the effect of solvents in extracting phenolic compounds and antioxidant activity. Thus, the aim of the present work was to evaluate different solvent systems and two carob varieties for the extraction of phenolic antioxidants from carob fruit. Then, the antioxidant effect of selected extracts were tested in four food systems in the search for new antioxidants for the food industry. Finally, High Performance Liquid Chromatography (HPLC) coupled on a-pre column 1,1-diphenyl-2-picrylhydrazyl (DPPH) assay was performed to find out the most powerful antioxidant compounds in carob fruit extracts.

## 2. Materials and Methods 

### 2.1. Chemicals and Fruit Material

(+)-Catechin, (−)-epicatechin, myricetin quercetin rutin, and syringic acid were obtained from Sigma-Aldrich (St Louis, MI, USA). Gallic acid, analytical reagents, and common solvents were provided by Scharlau (Barcelona, Spain).

Carob fruits (*Ceratonia siliqua* L., cvs ‘Tilliria’ and ‘Koumbota’) were harvested at the maturity stage from an experimental orchard (Avdimou, Limassol district, Cyprus; 34°41′15.4″ N, 32°45′34.8″ E). The carob fruits were pulverized by the employment of an electric coffee grinder (Bestron AKM1405 150W, Bestron Nederland BV, s-Hertogenbosch, The Netherlands). The moisture contents of Tilliria’ and ‘Koumbota’ carob fruits were 13.07 ± 0.89% and 13.65 ± 0.64%, respectively, as measured by a moisture analyzer (Kern MLS-50-3, KERN & Sohn GmbH, Balingen, Germany).

### 2.2. Preparation of Carob Fruit Extracts

The ultrasound-assisted extraction was applied for the extraction of phenolic antioxidants from carob fruit. Briefly, approximately 2.5 g of carob fruit were mixed with 25 mL of a single solvent or solvent system. Then, they were sonicated for 30 min at 30 °C in an ultrasound bath (UCI-50, 35 KHz, Raypa-R. Espinar, S.L., Terrassa, Spain). Samples were further centrifuged at 4700× *g*, 20 °C for 15 min. The supernatants were kept at −20 °C until analysis. In this study, water, methanol, ethanol, acetone, ethyl acetate, acidic methanol (methanol + water + HCl, 80:19:1; *v*/*v*/*v*), acidic ethanol (ethanol+ water + HCl, 80:19:1; *v*/*v*/*v*), acidic acetone (acetone + water +HCl, 80:19:1; *v*/*v*/*v*), methanol/water (80% and 50% *v*/*v*), ethanol/water (80% and 50% *v*/*v*), and acetone/ water (80% and 50% *v*/*v*) were used as extraction solvents.

For the determination of the antioxidant properties of extracts in model food systems, an amount of 15 g of carob fruit was extracted with 150 mL of an appropriate solvent or system solvent as described above. Finally, the solvents were removed using a rotary evaporator and freeze-dryer to obtain the dry extract.

### 2.3. Total Phenolic Content by High-Throughput Folin–Ciocalteu Assay

A microplate Folin–Ciocalteu assay was employed for the determination of the total phenolic content in carob fruit extracts [17]. More specifically, 50 μL of diluted extracts were mixed with 50 μL of Folin–Ciocalteu reagent (1:5, *v*/*v*) and 100 μL of sodium hydroxide solution (0.35 mol·L^−1^) in each well. After a period of 3 min, the absorbance of each sample was monitored at 760 nm. A standard curve of gallic acid was prepared and results expressed as mg gallic acid equivalents (GAE) 100 g^−1^ carob fruit.

### 2.4. Total Flavonoids by High-Throughput Technique

The determination of total flavonoids was carried out by mixing 100 μL distilled water, 10 μL of 50 g·L^−1^ NaNO_2_, and 25 μL of the sample or standard solution. After 5 min, 15 μL of 100 g·L^−1^ AlCl_3_ was added to the mixture. Finally, aliquots of 50 μL of sodium hydroxide solution (1 mol·L^−1^) and 50 μL of distilled water were added after 6 min. The mixture was shaken for 30 s in the plate reader prior to absorbance measurements at 510 nm [18]. A standard curve of catechin hydrate was prepared and the results expressed as mg catechin equivalents (CE) 100 g^−1^ carob fruit.

### 2.5. Antioxidant Activity by DPPH and Ferric Reducing Antioxidant Power (FRAP) High-Throughput Techniques

The 1,1-diphenyl-2-picrylhydrazyl (DPPH) assay measures the scavenging activity of antioxidants against colorful radicals of DPPH. The reaction mixtures consisted of 75 μL of extracts and 150 μL DPPH methanolic solution (0.2 mM). The mixtures were incubated for 30 min; then, the absorbance was read at 515 nm. Different concentrations of extracts were used for the determination and the antioxidant activity was expressed as EC_50_, plotting percent antioxidant activity against extract concentration [19].

The FRAP measurements were achieved by reacting 20 μL of the sample or standard compound with 180 μL FRAP solution (300 mM acetate buffer at pH 3.6, 10 mM 2,4,6-Tris(2-pyridyl)-*s*-triazine solution, and 20 mM ferric chloride solution in the ratio of 10:1:1, *v*/*v*/*v*) in 96-well plates. The reaction mixtures in well plates were allowed for 6 min at 37 °C and the absorbance was read at 595 nm. A standard curve of iron sulfate (FeSO_4_) was prepared and results expressed as μmol FeSO_4_ 100 g^−1^ carob fruit [20].

### 2.6. Antioxidant Effect of Carob Fruit Extracts in β-Carotene-Linoleate Model System

This protocol is based on the ability of the extracts to prevent the oxidative bleaching of colorful *β*-carotene in an emulsion. As described by Chandrasekara and Shahidi (2011), the carotene solution was prepared by dissolving 10 mg β-carotene in 10 mL of chloroform [21]. Then, 0.5 mL of the solution was transferred into a 50 mL round bottom flask and the solvent was removed with the employment of a rotary evaporator at 30 °C. After that, 20 mg of linoleic acid, 200 mg of Tween 20, and 50 mL of deionized water were mixed with vigorous shaking. In order to determine the *β*-carotene bleaching activity of the extract, 200 μL of emulsion were mixed with 20 μL of extracts in each well. The microplate was incubated at 45 °C and absorbance was monitored at 450 nm after 60 min. Gallic acid and catechin were also used as positive controls. The antioxidant activity was expressed as the percent inhibition relative to the control.

### 2.7. Antioxidant Effect of Carob Fruit Extracts in Sunflower Oil

Sunflower oil was enriched with carob extracts or pure compounds (control, 200 ppm carob fruit extracts, 200 ppm gallic acid, 200 ppm catechin). Five mL of the oil samples were transferred in 10 mL centrifuge vials and incubated at 60 °C in a shaker oven (Lab Companion SI-600R; Jeio Tech Inc., Seoul, Korea) for 10 days. After 0, 1, 3, 7, and 10 days, samples from each treatment were picked randomly to determine the thiobarbituric acid (TBA) value in triplicate. The TBA assay can measure secondary oxidation products of malonaldehyde [3]. More specifically, 200 mg of the oil samples were dissolved in 25 mL of 1-butanol. Subsequently, 5 mL of this solution was reacted with 10 mL of the TBA reagent (0.2%, *w*/*v*) and incubated for 2 h at 95 °C. After cooling, the absorbance was measured at 532 nm against a blank.

### 2.8. Antioxidant Effect of Carob Fruit Extracts in Oil-in-Water Emulsion

The antioxidative effect of extracts in a model food emulsion was performed according to previous work [22]. For the preparation of the emulsion, 5 g sunflower oil, 1 g Tween 20, and 50 mL of 20 mM sodium acetate buffer (pH 3.0) were homogenized with an Ultra-turrax (IKA^®^ Werke GmbH & Co.KG, Staufen, Germany) operated at 16,000× rpm. Then, the emulsions were fortified with carob extracts, gallic acid, and catechin at a final concentration of 200 mg·L^−1^ in the emulsion. A control without any antioxidant was also prepared. The emulsions were further homogenized with an ultrasonic processor operated for 5 min in an ice bath. Each emulsion was transferred into a plastic tube, capped, and incubated at 60 °C. After 0, 1, 2, and 3 days, samples from each treatment were picked randomly for the determination of the TBA value as described above.

### 2.9. Antioxidant Effect of Carob Fruit Extracts in Pork Model System

Ground pork was mixed with 20% (*w*/*v*) deionized water; carob fruit extracts as well as gallic acid and catechin were also added at the concentration of 300 mg·kg^−1^. After homogenization, the samples were cooked at 80 ± 2 °C for 40 min while stirring every 5 min. After cooling to room temperature, the samples were stored at 4 °C for 15 days [21]. After 0, 3, 7, 10, and 15 days, samples from each treatment were drawn for the determination of the TBA value. More specific, 2 g of each sample was mixed with 5 mL trichloroacetic acid (10% *w*/*v*) and vortexed at high speed for 2 min. Subsequently, 5 mL of TBA (0.02 M) was added and vortexed for 30 s. The samples were centrifuged at 3000× *g* for 10 min and the supernatants were heated in a boiling water bath for 45 min, cooled to room temperature in an ice bath, and the absorbance read at 532 nm.

### 2.10. Identification and Quantification of Phenolic Compounds Using HPLC

The chromatographic analysis was performed on a Waters HPLC system equipped with a vacuum degasser, binary pump, autosampler, thermostated column compartment, and dual λ absorbance detector (Waters Corporation, Milford, Ireland). The data collection and analysis was carried out by Empower software 2. An aliquot of carob extract (20 μL) was separated on a Waters Sherisorb^®^ ODS 2 (15 cm × 4.6 mm, 5 μm) column with a flow rate at 0.5 mL·min^−1^. The elution system consisted of solvents A (1 mL·L^−1^ acetic acid in water) and B (methanol). In particular, a gradient elution was applied as per our previous work described for the quantification of phenolic compounds in carob pulp extracts [8]. The identification and quantification of individual phenolic compounds was performed using standard compounds. More specifically, gallic acid, syringic acid, myricetin, quercetin and rutin, (+)-catechin, and (−)-epicatechin were studied in the present work.

### 2.11. HPLC Coupled on a Pre-Column DPPH Assay

One mL of freshly prepared DPPH methanol solution (13 mM) was mixed with 2 mL of carob extracts (10 mg·mL^−1^). The mixture was allowed to react at 25 °C in the dark for 30 min. Then, 20 μL of the mixture was injected into the HPLC and analyzed as described above. A blank control of carob extracts with the same volume of methanol was also analyzed [23].

### 2.12. Statistical Analysis

Software package SPSS v22.0 (SPSS Inc., Chicago, IL, USA) was used for the statistical analysis. The comparison of averages of each treatment was based on the analysis of variance (one-way ANOVA) according to Duncan’s multiple range test at a 5% significance level. 

## 3. Results and Discussion

### 3.1. Effects of Solvent Systems and Carob Variety on Phenolic Composition and Antioxidant Activity

The extraction of phenolic compounds from plant material is a well-studied research area; several extraction methods have been adopted for this purpose. More specifically, conventional methods such as solid–liquid and soxhlet extractions as well as modern methods such as microwave-assisted extraction, supercritical fluid extraction, and ultrasound-assisted extraction have been frequently used. Based on the literature, the ultrasound-assisted extraction is preferred for phenolics due to its simplicity and effectiveness [24,25]. Our previous study also demonstrated the increase in the yield of phenolics by using ultrasound irradiation [26]. This increase may be explained by enhancing the fragmentation process and promoting the release, diffusion, and dissolution of the components inside cells [24]. Taking into consideration that ultrasound assisted extraction does not require sophisticated equipment and can be transferred to large scale extraction, this method was selected in the present study.

Regarding the extraction system, the phenolics were recovered from plant matter using common alcohols (methanol, ethanol), acetone, diethyl ether, and ethyl acetate. However, the extraction of very polar phenolic acids cannot be achieved with the utilization of pure organic solvents, thus aqueous mixtures with alcohol and acetone are also recommended [27]. Furthermore, acidified organic solvents are used to improve the recovery of phenolics as they destroy cell membranes and dissolve some phenolics [28]. Although these recommendations can be a valuable guide for researchers, it is common sense that a solvent may be efficient on one plant and less efficient on another. Thus the efficiency of 14 solvent systems to extract phenolic antioxidants from carob fruit was studied through the determination of phenolic content, flavonoid content, DPPH, and FRAP assays.

The phenolic content of carob fruit extracts showed considerable variation as a function of the solvent systems used and ranged from 7.1 ± 0.9 mg GAE 100 g^−1^ to 382.0 ± 23.8 mg GAE 100 g^−1^. Results showed that the most powerful solvents to recover phenolics from carob fruits were water, acetone/water (50/50% *v*/*v*), acidic acetone, methanol and ethanol/water (50/50% *v*/*v*). In contrast, the pure organic solvents (ethyl acetate and acetone) yielded extracts with the lowest phenolic contents (Table 1). These findings are also supported by the activity coefficients predicted using the Universal Functional Group Activity Coefficient (UNIFAC) model in our previous study, highlighting the solubilization preference of natural phenols to alcohols and acetone [29]. Furthermore, aqueous mixtures with solvents usually extract higher phenolic contents than pure solvents as water may swell the plant material and increase extractability [30]. Surprisingly, the highest phenolic content was found in the aqueous extract due to the abundance of phenolic acids and gallotannins in the phenolic fraction [4]. The results also disclosed that the phenolic content of the extracts were not influenced by carob variety (*p* > 0.7).

The total flavonoid content of the carob extracts was also strongly affected by the extraction system as they fluctuated between 0 and 98.7 ± 2.4 mg CE 100 g^−1^. Results showed that the most suitable solvents for the recovery of carob flavonoids were acidic acetone and aqueous acetone mixtures. As many of the flavonoids occur in carob fruit as their glycoside derivatives, the use of acetone–water mixtures can be explained. On the other hand, the extracts of pure solvents (acetone, ethyl acetate, methanol, and ethanol) contained lower amounts of flavonoids. This means that the carob fruit comprises more glycosidic forms of flavonoids than aglycones. Indeed, the main flavonoids in carob fruit, namely quercetin, myricetin, and kaempferol, are found as glycosidic derivatives [31,32]. Finally, the comparison of carob varieties showed that the ‘Tilliria’ carob fruit was a slightly richer source of flavonoids than the ‘Koumbota’ carob fruit (*p* = 0.06).

The efficacy of solvents on the antioxidant activity of the extracts was also studied with DPPH and FRAP assays. These assays are based on different mechanisms to evaluate the antioxidant activity of carob extracts. The DPPH assay measures the ability of antioxidants to scavenge free radicals, whereas the FRAP assay counts the ability of antioxidants to donate electrons to the yellow ferric tripyridyltriazine complex. Moreover, a correlation between these assays and anti-lipid oxidation has been found for plant extracts [33]. Both assays demonstrated that the ethyl acetate and acetone extracts had the lowest antioxidant activity compared to the extract obtained using other solvents. The DPPH assay revealed that the most promising solvents for the recovery of carob antioxidants were acetone–water (80:20, *v*/*v*), acidic acetone, acidic methanol, and water. Regarding the FRAP assay, the use of pure water, ethanol–water (80:20, *v*/*v*), methanol, and acidic ethanol were recommended in order to obtain carob extracts with high antioxidant activity. Differences between the antioxidant assays can be ascribed to the different mechanisms of measurement. Furthermore, the solvent had a significant effect on the estimated antioxidant activity as it determined the rates and mechanism of reaction of the phenolics with reagents [34]. Similarly to total phenolic content, both carob varieties presented equal antioxidant properties.

Based on our findings, four solvents were evaluated for their potential to act as antioxidants in food model systems. More specifically, two pure solvents and two solvent systems were selected due to their ability to recover antioxidant phenolics. Water, acidic acetone, and acetone–water (80:20, *v*/*v*) were the most appropriate solvents to produce carob extracts, which are rich in phenolic antioxidants. The main drawback of these solvents is the co-extraction of sugars that are found in high concentrations in carob fruit. Thus, the pure methanol was also tested in food models due to its superiority against other pure organic solvents. Regarding the varieties, ‘Tilliria’ carob fruit was used for the preparation of extracts for the incorporation in food systems, although, all assays showed no significant differences between the two varieties. However, the ‘Tilliria’ variety is the most widespread in Cyprus and its exploitation of a carob antioxidant could attract the interest of the food industry.

### 3.2. Antioxidative Effect of Carob Extracts in Model Food Systems

The most promising carob extracts were tested as antioxidant additives in four model food systems that describe the requirements of food industry. In addition, the antioxidant potential of carob extracts was compared with the antioxidant potential of pure gallic acid and catechin as they are well-known antioxidants and found in carob fruit. First, the antioxidant activity of carob extracts was assessed by the β-carotene-linoleic acid system. This assay is based on the discoloration of β-carotene as it is extremely susceptible to free-radical mediated oxidation. Thus, the oxidative degradation products of linoleic acid induce the discoloration of β-carotene. Figure 1 demonstrates the inhibitory activity of free radicals generated during the peroxidation of linoleic acid. Catechin better inhibits the formation of dienes than gallic acid, as previously reported by [35]. Among the carob extracts, the acidic acetone extract had the highest antioxidant activity (70.3 ± 5.3%) followed by the acetone–water extract (62.1 ± 4.9%) and aqueous (52.4 ± 4.4%) and methanolic (50.1 ± 3.9%) extracts. The antioxidant activity of extracts were comparable with the corresponding activity of the pure compounds, highlighting the presence of other antioxidant constituents. Based on previous studies, carob flavonols such as quercetin, myricetin, and their derivatives are strong inhibitors of the formation of dienes in the β-carotene-linoleic acid emulsion. Their activity is attributed to the presence of the double bond between C-2 and C-3 and a free hydroxyl in the C-3 position [36]. Therefore, a chromatographic study of carob extracts is expected to confirm this hypothesis.

Next, the antioxidant efficiency of the carob extracts in sunflower oil in a water emulsion was determined. Figure 2 summarizes the effect of carob extracts on the inhibition of hydroperoxide formation in an oil in water emulsion during storage at 60 °C. After 24 h, a substantial increase in the concentration of hydroperoxides was monitored for the control samples. Results show that the carob extracts were effective in preventing hydroperoxide formation emulsion, whereas significant differences were found among the extracts. Aqueous and methanolic extracts did not inhibit lipid oxidation. In contrast, acidic acetone and acetone–water extracts had potent antioxidant activity in the emulsion system and was comparable to the pure catechin. The antioxidant activity of both extracts was higher than pure gallic acid, highlighting their antioxidant potential. Furthermore, it is well-known that gallic acid is the main constituent of the phenolic fraction, therefore other potent phenolic compounds are responsible for the strong inhibitory effect of these extracts. A similar response of carob extracts in the formation of hydroperoxides in an emulsion was recorded after 48 h storage at 60 °C. Finally, the antioxidant activity of acidic acetone and acetone–water extracts was equal with the activity of pure gallic acid and weaker than that of pure catechin after a period of 72 h. Overall, the acidic acetone and acetone–water extracts can be considered as antioxidative agents for sunflower oil in water emulsions.

Subsequently, the antioxidant activity of carob extracts was tested in an apolar medium such as sunflower oil. More specific, an amount of 200 ppm of carob extracts was incorporated in sunflower oil and their ability to prevent lipid oxidation for a period of 10 days was assessed. The hyperoxide contents were increased with the progress of the storage period. In the first three days, all carob extracts inhibited lipid oxidation and their potential was equal to that of the pure compounds (Figure 3a). The most remarkable findings may be found at the end of the storage period. Carob extracts were classified based on the ability to prevent lipid oxidation. In particular, the acetone–water extract demonstrated the highest antioxidant activity, followed by acidic acetone, methanol, and water extracts. Although the antioxidant activity of carob extracts was weaker than the antioxidant activity of the pure compounds, they can be considered as efficient inhibitors of oxidation in sunflower oil.

The cooked comminuted pork model system was also used to determine the potential of carob extracts as antioxidative additives. Results demonstrated a gradual increase in the hydroperoxide contents in cooked pork during storage at 4 °C for 15 days. Figure 3b clearly depicts a significant reduction in the formation of hydroperoxides when some carob extracts were added to the comminuted pork. More specifically, the acidic acetone and acetone–water extracts significantly inhibited the lipid oxidation during cold storage, although they were less potent than the pure compounds. The latter reduction is essential for food quality as lipid oxidation in muscle foods leads to the development of an off odor and off flavor in foods [21]. On the other hand, the aqueous and methanolic extract had no effect on the lipid oxidation of cooked pork. The different response of carob extracts in this food model is associated with the phenolic composition of carob extracts.

In conclusion, the utilization of different food model systems to evaluate the antioxidant activity of carob extracts revealed the most active extracts. The acidic acetone and acetone–water extracts significantly prevented the lipid oxidation in emulsions as well as in oil and pork. Thus, the study of the phenolic composition of extracts will allow us to understand the antioxidant behavior of extracts.

### 3.3. Determination of Antioxidant Phenolic Constituents of Carob Extracts

The carob extracts were subjected to HPLC analysis and the chromatograms revealed significant qualitative and quantitative differences among the studied extracts. More specifically, a reference HPLC fingerprint consisting of seven well-known phenolic antioxidants (gallic acid, syringic acid, catechin, epicatechin, quercetin, rutin, and myricetin) was developed to study the phenolic composition of the extracts. Figure 4 illustrates the phenolic composition of the carob extracts; the acidic acetone and acetone–water extracts are richer in phenolic compounds than that of the other two extracts. Gallic acid was the main phenolic compound in all extracts as its proportion to the phenolic content ranged from 63.7% to 74.6%. The methanol and water extracts also contained significant amounts of rutin; its concentration was 216.8 ± 15.6 μg·g^−1^ and 505.4 ± 18.0 μg·g^−1^, respectively. Myricetin and catechin were found at high concentrations in the acidic acetone and acetone–water extracts. In addition, the recognized phenolic antioxidants, quercetin and catechin, were detected only in these two extracts. All of these differences may be explained their strong antioxidant activity in food model systems.

In an attempt to decode the antioxidant activity of carob extracts, they were further analyzed with the use of HPLC coupled on a pre-column DPPH assay. The DPPH based antioxidant activity profile showed that several components decreased in the chromatogram of the reaction mixture of extracts and DPPH when compared to their original chromatograms. More specifically, the peak areas of all studied phenolic compounds significantly decreased after being reacted with the DPPH solution. The results showed that both flavonol aglycones, namely quercetin and myricentin, were the most potent antioxidant in carob extracts as they presented the highest rate of decrease (Table 2). Their presence only in the most active extracts could be responsible for their potent inhibitory effect in lipid oxidation. Gallic acid, the major component of extracts, also demonstrated a potent antioxidant potency as well as flavon-3-ols and rutin. Finally, a low contribution of syringic acid to the antioxidant activity of extracts was found according to the decreased rates of the peak areas. Our findings suggest that enrichment with quercetin and myricetin of carob extracts using fractionation and purification techniques is expected to further improve their antioxidant potential. Moreover, both flavonols have been recognized as potent lipid inhibitors [37,38].

## 4. Conclusions

The present work demonstrates that extraction solvents of different polarities yield carob extracts where the phenolic composition and antioxidant activity vary significantly. The present work provides a knowledge base for the selection of the most appropriate solvents to produce carob extracts for food applications. Based on these data, the potential of carob extracts to prevent lipid oxidation in food systems such as emulsions, sunflower oil, and cooked comminuted pork was explored and there are encouraging findings for the utilization of acidic acetone and acetone–water (50:50, *v/v*) extracts as antioxidant agents by the food industry. The fractionation and purification of these extracts and encapsulation may be a perspective to improve their antioxidant activity in food systems.

## Figures and Tables

**Figure 1 foods-09-00020-f001:**
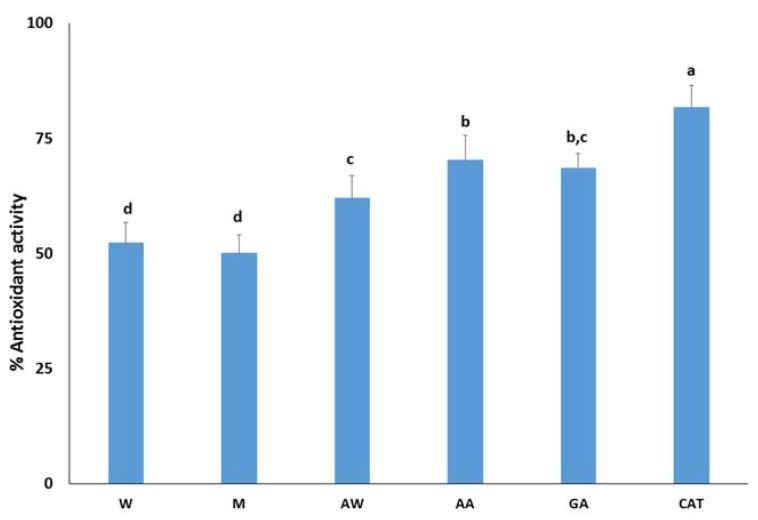
Antioxidant activity of carob extracts, gallic acid, and catechin in the β-carotene-linoleic acid system. The symbols W, M, AW, AA represent the water, methanol, acetone–water, and acidic acetone extracts; GA and CAT denote gallic acid and catechin, respectively. Means ± standard error labelled with the same letter did not differ significantly (*p* ≤ 0.05) according to Duncan’s test.

**Figure 2 foods-09-00020-f002:**
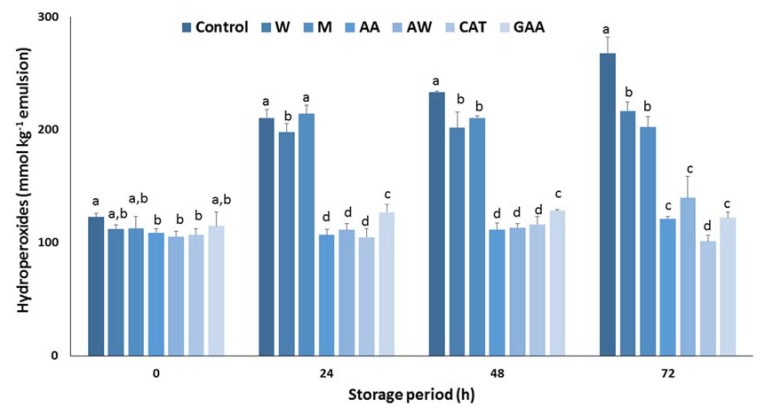
Antioxidant activity of carob extracts, gallic acid, and catechin in sunflower oil in water emulsion for 72 h at 60 °C. The symbols W, M, AW, AA represent the water, methanol, acetone–water, and acidic acetone extracts; GA and CAT denote gallic acid and catechin, respectively. Means ± standard error labelled with the same letter did not differ significantly (*p* ≤ 0.05) according to Duncan’s test.

**Figure 3 foods-09-00020-f003:**
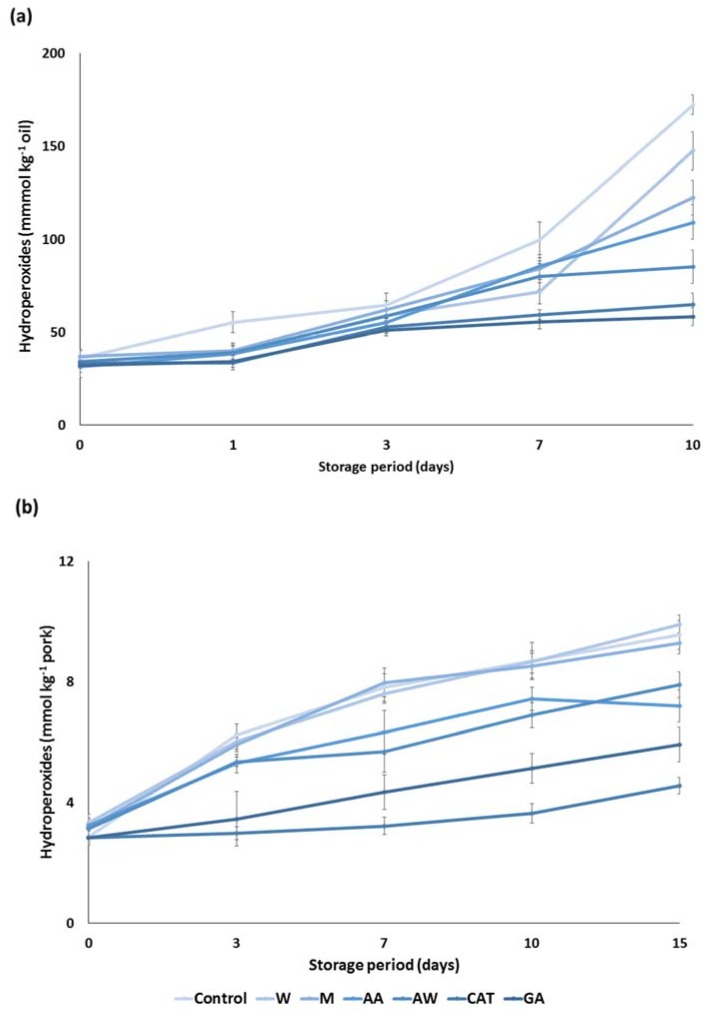
Antioxidant activity of the carob extracts, gallic acid, and catechin in (**a**) sunflower oil and (**b**) pork model system. The symbols W, M, AW, AA represent the water, methanol, acetone–water and acidic acetone extracts; GA and CAT denote gallic acid and catechin, respectively.

**Figure 4 foods-09-00020-f004:**
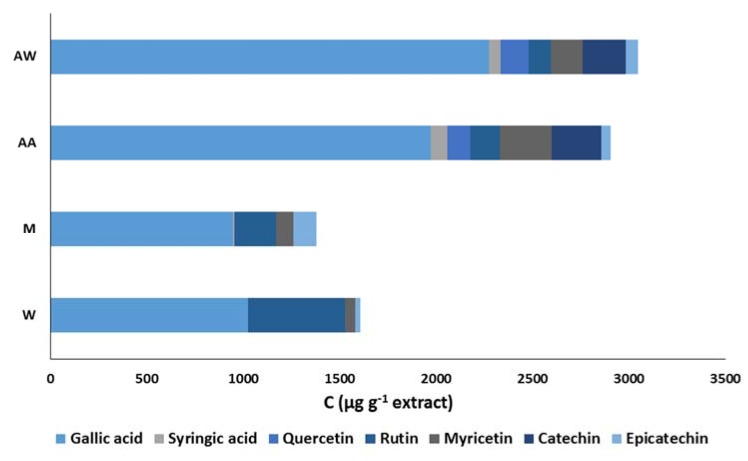
Comparison of the phenolic composition of carob extracts. The symbols W, M, AW, AA represent the water, methanol, acetone–water, and acidic acetone extracts, respectively.

**Table 1 foods-09-00020-t001:** Effect of extraction solvents on total phenolic content (TPC), total flavonoid content (TFC), and antioxidant activity of the carob extracts. Results are the means ± standard error. Labelled with the same letter did not differ significantly (*p* ≤ 0.05) according to Duncan’s test.

Solvent System	TPC (mg GAE^1^ 100 g^−1^)	TFC (mg CE^2^ 100 g^−1^)	DPPH IC_50_^3^ (mg mL^−1^)	FRAP (mg FeSO_4_ 100 g^−1^)
***‘Tilliria’ Carob Fruit***				
100% Water	382.0 ± 23.8 ^a^	22.9 ± 0.3 ^i,j^	1.9 ± 0.2 ^f,g,h,i^	339.7 ± 4.0 ^a^
100% Methanol	359.6 ± 13.5 ^a,b,c^	26.8 ± 3.7 ^h,i^	2.8 ± 0.1 ^e,f^	338.7 ± 39.3 ^a^
100% Ethanol	145.7 ± 19.0 ^k^	8.7 ± 0.1 ^l,m^	6.4 ± 0.9 ^b^	296.4 ± 19.8 ^a,b,c^
100% Acetone	80.3 ± 6.9 ^l^	Not detected	> 20 ^a^	98.8 ± 5.5 ^j^
100% Ethyl acetate	7.1 ± 0.9 ^m^	5.7 ± 1.0 ^m,n^	> 20 ^a^	29.2 ± 4.1 ^k^
Methanol + water + HCl (80:19:1, *v*/*v*/*v*)	288.4 ± 19.4 ^h^	70.5 ± 1.8 ^c^	1.6±0.2^g,h,i^	245.5 ± 28.1 ^c,d,e,f^
Ethanol + water + HCl (80:19:1, *v*/*v*/*v*)	315.8 ± 6.7 ^e,f,g,h^	55.7 ± 5.7 ^d,e,f^	2.1±0.4 ^e,f,g,h,i^	270.2 ± 7.4 ^b,c,d^
Acetone + water + HCl (80:19:1, *v*/*v*/*v*)	348.6 ± 5.8 ^a,b,c,d,e^	98.7 ± 2.4 ^a^	1.4 ± 0.0 ^h,i^	271.6 ± 29.1 ^b,c,d^
Methanol + water (80:20, *v*/*v*)	342.2 ± 9.4 ^b,c,d,e^	49.3 ± 1.4 ^f^	2.2 ± 0.5 ^e,f,g,h,i^	205.7 ± 8.9 ^f,e,g,h^
Methanol + water (50:50, *v*/*v*)	362.0 ± 5.9 ^a,b^	51.2 ± 0.1 ^f^	1.9 ± 0.2 ^e,f,g,h,i^	257.8 ± 31.8 ^c,d,e^
Ethanol + water (80:20, *v*/*v*)	304.2 ± 3.4 ^f,g,h^	22.7 ± 2.7 ^i,j^	2.9 ± 0.2 ^e,f,^	335.6 ± 0.6 ^a^
Ethanol + water (50:50, *v*/*v*)	323.9 ± 3.0 ^c,d,e,f,g,h^	52.8 ± 0.9 ^e,f^	2.1 ± 0.3 ^e,f,g,h,i^	195.5 ± 3.3 ^e.g,h^
Acetone + water (80:20, *v*/*v*)	328.6 ± 2.0 ^b,c,d,e,f,g^	88.6 ± 9.0 ^b^	1.6 ± 0.2 ^i^	224.2 ± 9.1 ^d,e,f,g,h^
Acetone + water (50:50, *v*/*v*)	347.2 ± 1.8 ^a,b,c,d,e^	77.2 ± 6.7 ^c^	1.4 ± 0.1 ^h,i^	227.1 ± 27.8 ^d,e,f,g^
***‘Koumbota’ Carob Fruit***				
100% Water	351.6 ± 16.2 ^a,b,c,d,e^	27.1 ± 1.7 ^h,i^	2.0 ± 0.4 ^e,f,g,h,i^	324.1 ± 37.1 ^a,b^
100% Methanol	323.0 ± 10.3 ^c,d,e,f,g,h^	10.3 ± 1.7 ^l,m^	2.4 ± 0.2 ^e,g^	259.1 ± 36.9 ^c,d,e^
100% Ethanol	204.9 ± 8.6 ^j^	3.7 ± 0.7 ^m,n^	4.3 ± 0.2 ^c^	209.9 ± 37.7 ^f,e,g,h^
100% Acetone	57.9 ± 3.5 ^l^	15.5 ± 2.0 ^j,k,l^	> 20 ^a^	79.4 ± 1.4 ^j,k^
100% Ethyl acetate	17.4 ± 4.3 ^m^	16.3 ± 1.7 ^j,k,l^	> 20 ^a^	41.1 ± 6.5 ^k^
Methanol + water+ HCl (80:19:1, *v*/*v*/*v*)	343.2 ± 33.7 ^b,c,d,e^	38.8 ± 3.0 ^g^	1.8 ± 0.2 ^f,g,h,i^	225.5 ± 38.3 ^d,e,f,g,h^
Ethanol + water+ HCl (80:19:1, *v*/*v*/*v*)	320.8 ± 28.6 ^d,e,f,g,h^	33.4 ± 4.8 ^g,h^	2.3 ± 0.4 ^e,f,g,h,i^	320.8 ± 35.3 ^a,b^
Acetone + water+ HCl (80:19:1, *v*/*v*/*v*)	337.9 ± 5.5 ^b,c,d,e,f^	86.1 ± 5.3 ^b^	1.5 ± 0.1 ^g,h,i^	223 ± 22.8 ^d,e,f,g,h^
Methanol + water (80:20, *v*/*v*)	228.4 ± 25.8 ^i,j^	20.8 ± 1.6 ^i,j,k^	3.0 ± 0.4 ^d,e^	170.3 ± 10.4 ^g,h^
Methanol + water (50:50, *v*/*v*)	293.2 ± 32.0 ^g,h^	18.4 ± 2.2 ^j,k^	2.6 ± 0.1 ^e,f,g,h^	187.9 ± 16.8 ^e,f,g,h^
Ethanol + water (80:20, *v*/*v*)	255.0 ± 20.0 ^i^	13.9 ± 1.5 ^k,l^	4.0 ± 0.6 ^c,d^	281.8 ± 28 ^a,b,c,d^
Ethanol + water (50:50, *v*/*v*)	355.5 ± 22.9 ^a,b,c,d^	32.1 ± 3.2 ^g,h^	2.6 ± 0.3 ^e,f,g,h^	167.0 ± 3.1 ^h^
Acetone + water (80:20, *v*/*v*)	321.7 ± 11.2 ^c,d,e,f,g,h^	59.4 ± 3.2 ^d,e^	1.5 ± 0.0 ^g,h,i^	188.2 ± 24.6 ^e,f,g,h^
Acetone + water (50:50, *v*/*v*)	351.4 ± 4.1 ^a,b,c,d,e^	62.56 ± 6.1 ^d^	2.6 ± 0.1 ^e,f,g^	325.5 ± 27.7 ^a,b^

GAE: gallic acid equivalents; CE: catechin equivalents; EC_50_: the concentration of the extracts required for 50% of the antioxidant activity. The same letter in each column are not significantly different at *p* ≤ 0.05, according to the Duncan’s test.

**Table 2 foods-09-00020-t002:** Decreased rate of peak areas in selected phenolic compounds in carob fruit extracts.

	Hydroxybenzoic Acids		Flavonols			Flavan-3-ols	
Extract	Gallic Acid	Syringic Acid	Quercetin	Rutin	Myricetin	Catechin	Epicatechin
**Water**	80.08	nd *	nd	67.09	94.53	nd	77.33
**Methanol**	78.29	67.16	nd	72.37	92.68	nd	70.58
**Acidic Acetone**	74.66	56.64	89.66	82.36	85.99	79.54	79.08
**Acetone Water**	71.89	51.25	88.91	80.55	87.33	77.63	76.44

* nd: not detected.

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
