# Peer review of "Utilization of Carob Fruit as Sources of Phenolic Compounds with Antioxidant Potential: Extraction Optimization and Application in Food Models"

_foods, 2019, doi:10.3390/foods9010020_

Round 1
Reviewer 1 Report
Comments to the Authors:
The authors of this paper present an interesting preliminary study on the Utilization of carob fruit as sources of phenolic antioxidants: extraction optimization and application 3 in food models. Nevertheless, this work might be improved taking into account the following suggestions:
During the initial checking of the manuscript, we saw that there are few overlaps in your manuscript. That means the highlighted parts( see attached Turnitin file) is similar to some published works. Please kindly revise/refresh these parts in your manuscript. Abstract needs to rewrite up to 200 words (according to the journal instructions) and should show quantitative relevant findings of the study.
COMMENT:
Please correct the free radical DPPH∙ as 2 DPPH∙ in all manuscript. Page 2, paragraph 2.1: All the methods are well described. However, some experimental details still need to be added and justified: The selection of extraction conditions (ratio, time , temperature) should be commented/discussed. Furthermore, the humidity percentage of the carob fruits had been measured. Please explain why in the DPPH∙ scavenging activity methodology the incubated performed also in 60minutes. Line 157 (kg-1) please correct as (kg -1) Please correct the word “activitity” in line 338 and “activiti” in line 359. All figures need to improve in order to be legible.

Author Response
Please find attached a point-by-point reply on comments of reviewer.

Reviewer 2 Report
Comments and Suggestions for Authors
The paper entitled “Utilization of carob fruit as sources of phenolic antioxidants: extraction optimization and application in food models”. The aim of this work was to evaluate different solvent systems and two carob varieties for the extraction of phenolic antioxidants from carob fruit. This paper is very interesting and useful. If the article is adopted, it needs improvement.
It is mandatory to correct the manuscript in some points:
# I suggest change the title on “Utilization of carob fruit as sources of bioactive compounds and antioxidants activity: extraction optimization and application in food models”
# line 23: “HPLC and off-line DPPH-HPLC” this names should be define
# line 18: should be “Subsequently, the most active extracts (water, methanol, acidic acetone, and acetone-water)…”
#line 64: should be „off-line HPLC-DPPH” not „on-line HPLC-DPPH”?
#line 68: should be „(-)-Epicatechin”
#line 69: Could you define „MO”
#line 71-72: please provide geographical data.
#line 91 and 98: „high-throughput”? What does it mean? This methods are basic to tested total of polyphenols and flavonoids.
#line 113: Could you define „FRAP”
#line 182: should be „An aliquot of 1 mL of freshly prepared DPPH˙ methanol solution (13 mM) was mixed with 2 mL carob extracts…”
#Table 1 all results should be one decimal place. Correct.
#line 297: should be „After 24 h, a”
#line 304: should be „…after 48 h storage at 60 °C.”.
#line 311: should be „…72 h at 60 °C…”
#Fig. 3 are not legible.
# line 419: „Waste and Biomass Valorization” should be Abbreviated Journal Name
Author Response

(The authors gave the same response as above.)

Round 2
Reviewer 2 Report
All suggestion have been correction. The article can now be published.